# Physical Properties of Betaine-1,2-Propanediol-Based Deep Eutectic Solvents

**DOI:** 10.3390/polym14091783

**Published:** 2022-04-27

**Authors:** Qicheng Chen, Nan He, Jing Fan, Fenhong Song

**Affiliations:** School of Energy and Power Engineering, Northeast Electric Power University, Jilin City 132012, China; 20142551@neepu.edu.cn (Q.C.); 20152613@neepu.edu.cn (N.H.); fenhongsong@neepu.edu.cn (F.S.)

**Keywords:** deep eutectic solvents, betaine, 1,2-propanediol, physical property

## Abstract

Due to their splendid advantages, deep eutectic solvents have attracted high attention and are considered as analogues of ionic liquids. Deep eutectic solvents (DESs) are homogeneous mixtures formed by two or three green and cheap components through hydrogen bond, which is divided into hydrogen bond acceptors (HBA) and hydrogen bond donors (HBD). Recently, Betaine has been widely used as a hydrogen bond acceptor. In this work, four DESs were synthesized by blending betaine as HBA and 1,2-propanediol as HBD in four molar ratios (1:3.5, 1:4, 1:5, 1:6). Then, the physical properties of these DESs were measured. The density values were measured within the temperature range (293.15 K to 363.15 K) at atmospheric pressure, whereas the surface tension and viscosity data were determined in four and seven temperatures between 293.15 K and 353.15 K. The relationship between the density and surface tension with temperature have been analyzed and have been fitted as a linear function. The commonly used Arrhenius model was used to describe the dependence between viscosity and temperature. The results of this study are important not only for the DESs’ industrial applications but also for the research on their synthesis mechanism and microstructure.

## 1. Introduction

With the introduction of the concept of green chemistry, many scholars have explored ionic liquids more and more deeply. Due to their remarkable properties (low vapor pressure, high boiling points, low flammability, excellent dissolution ability, thermal stability and chemical stability, wide liquid range, and so on), ionic liquids are treated as a green solvent and have been used in the fields of separation technology, biocatalysis, organic synthesis, and electrochemistry. Currently, there are about 250 ionic liquids that have been commercialized. However, it has been reported that some defects, such as complex synthesis, purification difficulties, high price, and poor biodegradability, limit the large-scale industrial application of ionic liquids. Therefore, it is necessary to develop green solvents that can maintain the excellent physical properties of ionic liquids while overcoming their disadvantages [1,2,3,4].

With the improvement of solvent performance requirements, deep eutectic solvents (DESs) are recognized as ionic liquid analogues, have attracted high attention, and have become another novel class of green solvent [5,6,7,8,9]. Since Abbott’s team discovered and named the deep eutectic solvent in 2003 [10], DESs have been extensively studied. It has been found that DESs can not only overcome the shortcomings of organic solvents, such as volatility and high melting point of ionic liquids, but also maintain the advantages of these two types of reagents. DESs are widely used in many fields such as electrochemistry, separation technology, organic synthesis, and so on because of their simple preparation, low price, non-toxicity, low vapor pressure, and wide electrochemical window. It is confirmed that DESs have a very broad application prospect [11,12,13,14,15].

DESs are homogeneous mixtures that are made by blending two or three hydrogen bond donors (HBD) and one hydrogen bond acceptor (HBA) with the melting point lower than each of the components. Ammonium salt, amino acids, and metal chloride are usually used as HBA, whereas amines, amides, carboxylic acids, alcohols, and amino acids are employed as HBD. In the early studies, cholinium chloride (choline) was the most commonly used compound to prepare DESs [10,11,16]. To expand the application of DESs, it is necessary to research new HBAs, which can be substituted for choline chloride, that are cheaper, more readily available, and non-toxic. In this context, betaine has gradually entered the scope of research scholars. Betaine is a kind of inner salt consisting of an anionic carboxylic acid group and a positively charged quaternary nitrogen [17]. In theory, betaine cannot be used to prepare DES because the betaine molecule itself should establish a strong anion-cation interaction. After all, the anion-cation interaction is much stronger than the hydrogen bond formed by HBA and HBD. However, in practice, the positively charged quaternary nitrogen in betaine is highly shielded by methyl groups, thereby inhibiting the electrostatic interaction with anions [18]. Therefore, betaine has been considered a suitable HBA for the synthesis of DES in recent years due to its own structural characteristics and its sustainability. 

The betaine-based DESs are being used in the fields of extraction, separation, CO_2_ adsorption, catalysts, nanomaterials, and so on. For example, Li et al. [19] synthesized six kinds of betaine-based DESs and successfully applied them in the extraction of protein from calf blood, finding that salt concentration played an important role in the optimal extraction conditions. Fanali et al. [20] reported that nutraceutical compounds were extracted from coffee grounds more efficiently using betaine-based DESs than the DESs using choline chloride as HBD. Additionally, three deep eutectic solvents based on betaine, glycerol, ethylene glycol, and propylene glycol were prepared by Kučan’s research group and successfully used as selective solvents for extracting only nitrogen compounds from gasoline [21]. Mojtabavi et al. [22] used a betaine-based DES to increase the enzyme stability at various temperatures and pH levels as an appropriate alternative strategy to improve the Laccase stability. Ribeiro et al. [23] found that DESs formed from betaine hydrochloride, organic acids, polyols, and amides were an alternative as non-aqueous media for enzymatic reactions of lipases. Additionally, lipase showed great thermostability and relative activity in the presence of betaine hydrochloride and urea (molar ratio 1:4) relative to aqueous solution. Moreover, betaine-based DES has a strong ability to absorb carbon dioxide [24], and another report by Crescenzo’s group showed that gold nanoparticles could be synthesized from betaine-based DESs without any surfactant or capping agent [25]. Although a growing number of researchers have reported the applications of some betaine-based DESs, the physical properties of betaine-based DESs are still poorly studied. However, especially in the design of the optimal system for a given application, it is important to strengthen the exploration of the physical properties of DESs, such as density, surface tension, and viscosity, among others [26,27,28,29,30]. 

Within this background, in this work, four DESs mixed using betaine as the HBA and 1,2-propanediol as the HBD in different molar ratios have been synthesized, and their physical properties were measured, including density (from 293.15 K to 363.15 K), surface tension (from 293.15 K to 353.15 K), and viscosity (from 293.15 K to 353.15 K). The changes in density, surface tension, and viscosity of DESs in relation to temperature were obtained, and the corresponding dependencies were fitted. Meanwhile, the effects of different molar ratios on the density, surface tension, and viscosity of DESs were summarized.

## 2. Materials and Methods

### 2.1. Materials

Betaine (CAS: 107-43-7) and 1,2-propanediol (CAS: 57-55-6) were purchased from Aladdin Chemistry Co. Ltd. (Shanghai, China) and were used directly without further purification with the mass fraction purity better than 99% from supplier (Table 1). 

### 2.2. Preparation of DESs

Four DESs were composed in the four molar ratios (1:3.5, 1:4, 1:5, 1:6) using betaine and 1,2-propanediol (as shown in Table 2). The weighed samples were shaken by an incubator at 300 rpm and 353.15 K for 2 h until a colorless liquid was obtained. Four DESs were prepared at atmospheric pressure and under tight control of moisture content. Then, the synthesized samples were put in a desiccator for at least 24 h. In addition, we tried to synthesize another two DESs with the molar ratios of betaine to 1,2-propanediol being 1:2 and 1:3. However, it failed to compose the homogeneous mixture in these ratios at atmospheric pressure.

### 2.3. Physical Properties Measurement

An Anton Paar digital densimeter (Model: DMA 5000M), which is based on the vibrating U-tube method, was used to measure the DESs’ density. The densimeter can be used to conduct density measurements in temperatures ranging from 273.15 K to 363.15 K, and its repeatability was 0.000001 g/cm³ from supplier. The density measurement’s relative standard uncertainty was specified as 0.001. Additionally, the densimeter must be well-cleaned using acetone or ethanol and dried before and after the measurements [31]. 

The measurement of surface tension was conducted by the experimental system that has been established in our laboratory based on the two capillaries rise method. The capillary rise method, which is one of the most accurate methods to determine the surface tension, not only has a relatively complete theory but also can strictly control the experimental conditions, and its principle has been introduced in details elsewhere [32,33]. In this work, the uncertainty of surface tension measurement was better than ±0.2 mN·m^−1^.

The viscosity of DESs were measured by an accurate viscosity measurement system that was composed of a Brookfield rheometer (Model: RST-CC), a Brookfield water thermostatic (Model: TC-550SD), and other accessories. Rheological evaluation through controlled stress and controlled rate measurements offers superior viscosity profiling, thixotropic response, yield stress determination, and creep analysis. The measurement system can be used to measure the viscosity in temperatures ranging from 253.15 K to 423.15 K with an uncertainty of ±0.04 K. 

## 3. Results and Discussion

### 3.1. Density

Density is a key physical property of materials and has an impact on mass transfer or chemical processes, and it also can provide information about the intermolecular interactions in DESs. Generally, the densities of DESs reported in the literature are higher than that of water. In this work, four prepared DESs were measured in the temperature range (293.15–363.15) K. During the experimental measurement, the density in the same temperature was measured in triplicate to obtain the average data of the density. Table 3 gives the density of DESs and pure 1,2-propanediol in different temperatures. It should be noted that the density of the DES decreases with increasing the molar ratio of 1,2-propanediol. Densities of four DESs decreased as follows: DES1 > DES2 > DES3 > DES4, ranging from 1.075588 to 1.062906 kg·m^−^^3^ at 293.15 K and from 1.02875 to 1.013525 kg·m^−^^3^ at 363.15 K. In addition, the density of all DESs is higher than that of the pure 1,2-propanediol (for example, 1.0362 kg·m^−^^3^ at 293.15 K) at the same temperature. As most betaine/polyol-based DESs are denser than water, as described by Zhang [12]. Such measurements were consistent with the density changes of DESs synthesized using betaine and levulinic acid in different ratios reported by Mulia [22], and the addition of salt increases the density of the isolated HBD. However, the density of the mixture of betaine and glycerol measured by Rodrigues was less than that of the isolated glycerol [34]. It is noted that in some cases, selecting the appropriate composition and molar ratio of HBA to HBD provides a method for changing the density of the eutectic mixture. This phenomenon may be explained in terms of free volume [35]. Figure 1 displays the relationship between the DESs’ density and temperature. Clearly, the density of all DESs linearly decreases with temperature rise. This is due to the higher molecular activity and the enhanced mobility of the molecules as the temperature increases, thereby increasing the molar volume of the solution and ultimately leading to a decrease in density [36].

The experimental data of the density were fitted by a linear function of temperature:*ρ* = *A* + *BT*,(1)
where *ρ* is the density, *T* is the temperature, and *A* and *B* are two constants. The values of *A* and *B* of different DESs are presented in Table 4.

### 3.2. Surface Tension

Surface tension is a useful physical property which is needed in a lot of areas, such as emulsions and surfactants applications. Measuring the surface tension of DES can be used as a method to understand the changes of DES molecular environment caused by the changes of composition and temperature. The vast majority of DES surface tension measurements reported so far are based on choline chloride and have found that the surface tension of choline-based DES decreases with the increase in temperature [29,37]. In addition, Gajardo-Parra [38] measured the surface tension of DES composed of choline chloride as HBA and ethylene glycol, phenol, and levulinic acid as HBD at 298.15 K and 101.3 kPa. The reported surface tension of choline chloride and ethylene glycol (45.66 mNm^−^^1^ at 298.15 K and 101.3 kPa) was lower than that of the pure ethylene glycol (48.90 mNm^−^^1^ measured). Very few studies reported the surface tension of betaine-based DESs. In this study, the measurement of surface tension of the studied betaine-based DESs in temperatures ranging from 293.15 K to 353.15 K was conducted. Table 5 lists the experimental surface tension results of solvents, and they are also higher than the pure 1,2-propanediol. Figure 2 and Figure 3 show these data graphically. It can be seen clearly that the surface tension of the four prepared DESs decreases as the temperature increases, which is consistent with the variation trend of the surface tension of choline-based DES. It is worth noting that the surface tensions of betaine-based DESs measured in this work are all higher than those of the pure 12-propanediol at the same temperature. In this study, a reproducible experiment was carried out in order to obtain the measurement results, indicating that the measurement results in this work have certain credibility. It can be concluded that the surface tension of betaine-based DES is higher than that of pure substance, whereas choline chloride-based DES has a lower surface tension than pure substance. These findings might depend on the intermolecular forces and hydrogen bonding between HBA and HBD; thus, further studies need to be conducted to investigate these from a molecular level.

From Figure 3 it can be seen that the surface tension of the four DESs first decreased and then increased from DES1 to DES4. DES1 has the highest surface tension, whereas DES3 has the lowest surface tension at the same temperature. The reason for this is that the HBD and HBA mixed at the appropriate molar ratio can generate the optimum hydrogen bonding interaction force, and the stronger the hydrogen bonding force appears, the smaller the DESs’ surface tension is [39]. 

The experimental results of the density and surface tension were fitted as a linear function of temperature as follows:*σ* = *a* + *bT*,(2)
where *σ* is the surface tension, *T* is the temperature, and *a* and *b* are two constants. The values of *a* and *b* of different DESs are presented in Table 6.

### 3.3. Viscosity

Viscosity is a physical quantity that measures the amount of frictional resistance between adjacent fluid layers when a fluid flows. It is well known that DESs generally have a higher viscosity, similar to most of the ionic liquids. The high viscosity of DES has brought inconvenience to its practical application and is regarded as the main obstacle to its widespread use. Therefore, viscosity is another important property of solvents that must be addressed. In this work, the viscosity of prepared DESs and pure 1,2-propanediol in the temperature range (293.15–353.15 K) was measured, and the experimental results are listed in Table 7. Figure 4 shows the trend of the viscosity with increasing temperature. It can be seen that the viscosity of DESs is greatly influenced by the temperature. The viscosity decreased significantly with increasing temperature, especially in the low temperature range. In general, the increase in temperature leads to the weakening of the intermolecular force in the liquid and the acceleration of the average movement speed of the molecules, thereby improving the kinetic energy of the molecules, promoting the flow between the molecules, and resulting in the decrease in dynamic viscosity of the liquid. 

Figure 4 also shows that for a given temperature, the viscosity of the four prepared DESs are in the following descending order: DES1 > DES2 > DES3 > DES4. Although the viscosity of DES decreased with increasing molar ratio of 12-propanediol, the viscosity of all DESs increased compared with that of the isolated 1,2-propanediol. The main reasons are that, on the one hand, there are a large number of hydrogen bond networks between each component of HBA and HBD, which leads to the lower mobility of free species within DES [12]; on the other hand, it is due to the thickening potential of betaine. Consequently, in some practical applications that require lower viscosity, selecting the appropriate ratio of HBA to HBD or preheating and increasing the temperature are very simple and effective techniques to reduce viscosity.

The Arrhenius model and Vogel-Fulcher-Tammann (VFT) behavior are most commonly used to describe the temperature dependence of DES viscosity [40]. Therefore, the Arrhenius model is considered to be more accurate and simple. The measured viscosities could be described by the Arrhenius model over the studied temperature range as shown: (3)$$\ln\eta =\ln\eta_{0}+\frac{E_{\eta }}{RT},$$
where *η* stands for the viscosity, *η*_0_ is a constant, *E_η_* refers to the activation energy, *R* is the gas constant, and *T* is the temperature. Figure 5 indicates that the ln*η* − *T*^−1^ relationship tends to be linear. According to the linear fitting, the values of *η*_0_ and *E_η_* are given in Table 8.

Then we calculated the viscosity of DESs according to the Arrhenius model at different temperatures and compared them with the experimental data. The deviation values were also calculated, as listed in Table 9. It can be seen that the deviation is significant between the calculated and experimental data. The Arrhenius model is famous for expressing the viscous behavior of DESs, but it is more suitable for predicting the viscosity of liquids at high temperatures or for measuring viscosity over a narrow temperature range [40]. The temperature range (293.15–353.15 K) we used to measure the viscosity of DESs is a little large, resulting in a large deviation between the experimental values and the calculated values.

## 4. Conclusions

DESs have been considered to be green solvents due to their remarkable solubilizing power and have been recommended for industrial applications. In this study, four DESs were successfully prepared by using betaine as the HBA and 1,2-propanediol as the HBD in different molar ratios. The physical properties (density, surface tension, and viscosity) of these four DESs were determined and reported in the large temperature range. The results show that the density, surface tension, and viscosity of these DESs decrease with the temperature increasing and are higher than the isolated 1,2-propanediol. The density and viscosity of the DESs showed a linear relationship with temperature, whereas the dependence of viscosity and temperature can be obtained using the Arrhenius model. In addition, with the increase in the molar ratio of 1,2-propanediol, the density and viscosity of DES decreased, and the surface tension showed a trend of first decreasing and then increasing. The molar ratio 1:5 of betaine and 1,2-propanediol had the lowest surface tension. It is believed that the results in this study would be important for utilization of DESs in practical industrial applications.

Although some problems need to be further studied, such as the issue that homogeneous mixtures could not be formed for HBA: HBD ratios 1:2 and 1:3, the effect of betaine on the surface tension of DESs is unclear. Molecular dynamic simulation provides a useful tool to investigate the molecular interaction between different HBA and HBD. Additionally, molecular perspective investigations conducted by MD simulations might be helpful to understand the formation mechanism of DESs and the mechanism behind the optimal hydrogen bonding. 

## Figures and Tables

**Figure 1 polymers-14-01783-f001:**
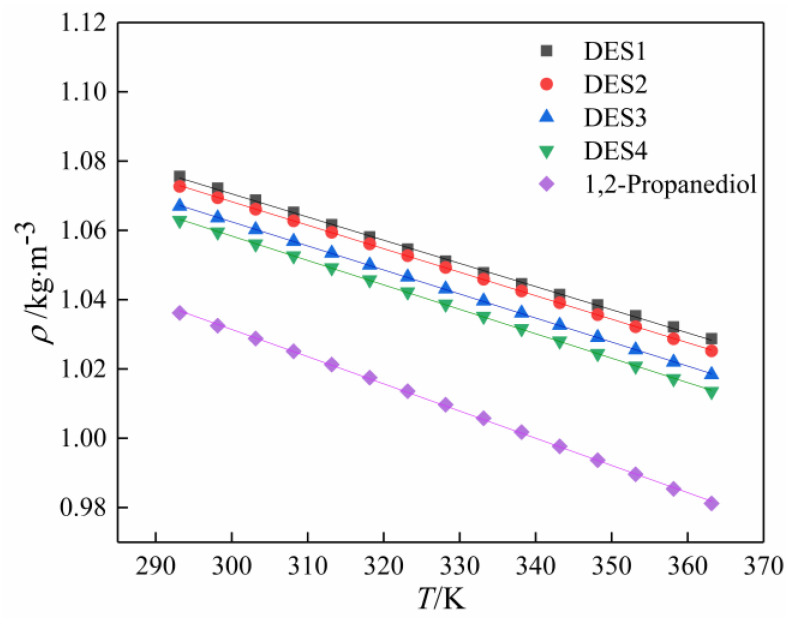
Temperature dependence of density of 1,2-propanediol and DESs.

**Figure 2 polymers-14-01783-f002:**
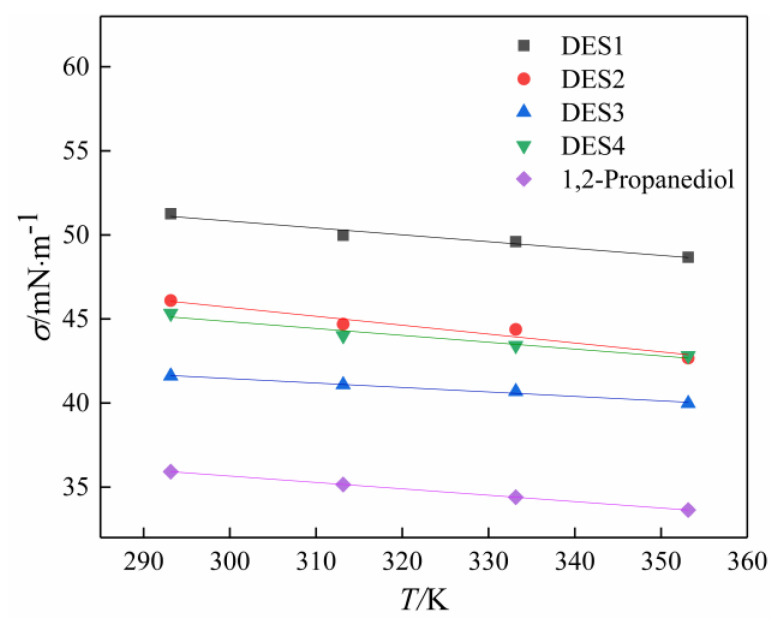
Plot of *σ* vs. T of 1,2-propanediol and four DESs.

**Figure 3 polymers-14-01783-f003:**
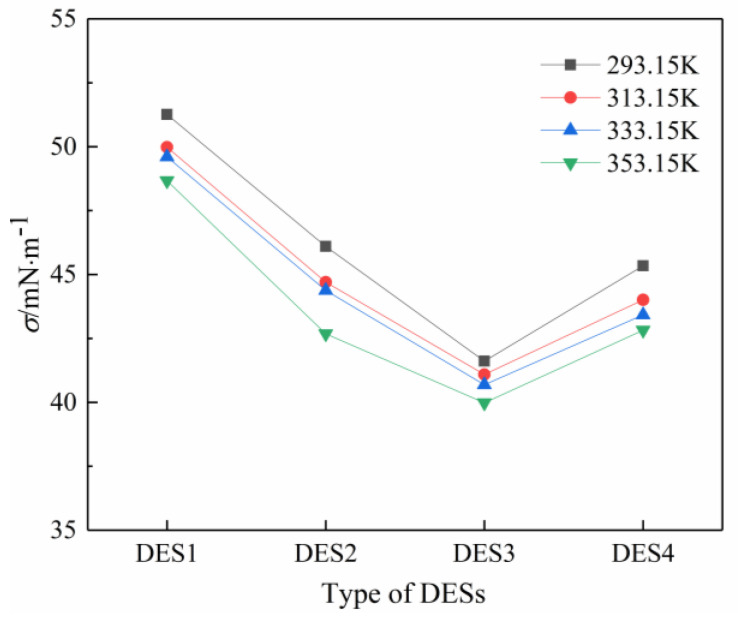
Variations of surface tension with type of DESs in different temperatures.

**Figure 4 polymers-14-01783-f004:**
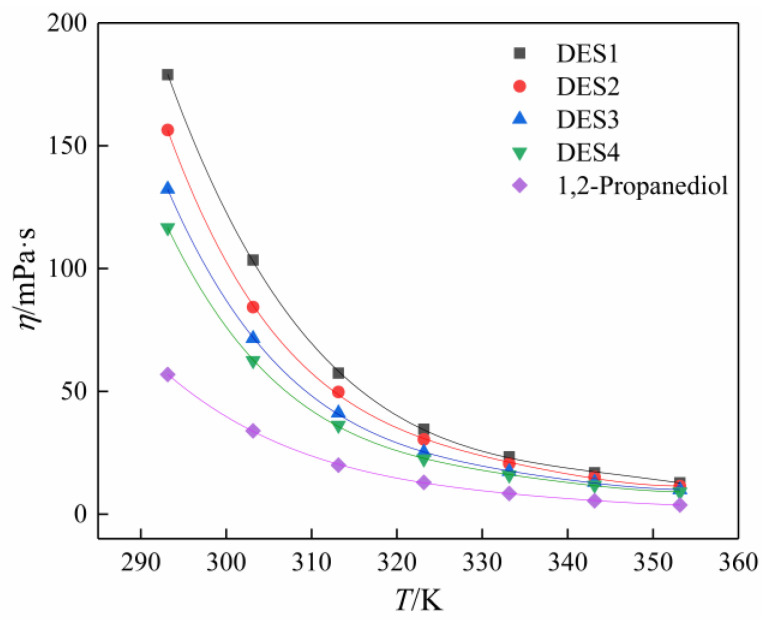
Variations of viscosity with temperature.

**Figure 5 polymers-14-01783-f005:**
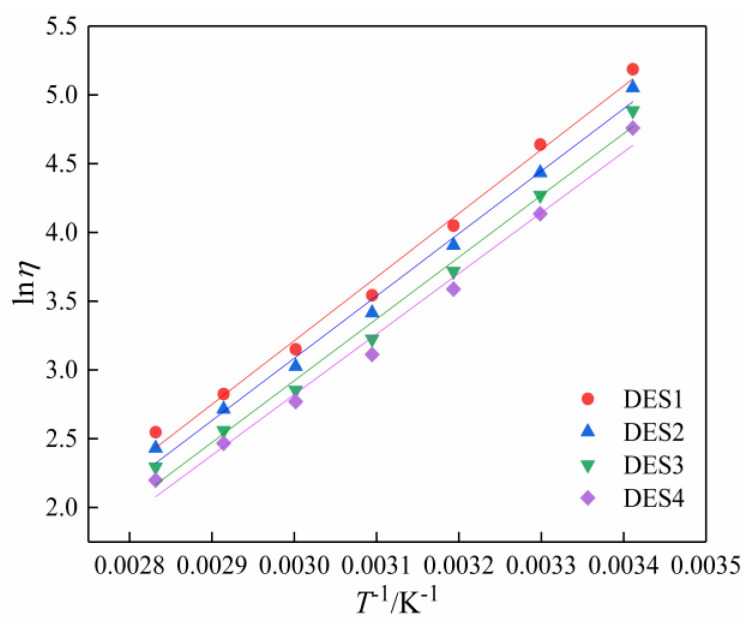
Relationship between ln*η* and *T*^−1^ of different DESs.

**Table 1 polymers-14-01783-t001:** Chemical specifications.

Chemicals	CAS Number	Chemical Formula	Source	Mass Fraction Purity (Supplier)
Betaine	107-43-7	C_5_H_11_NO_2_	Aladdin	>0.99
1,2-Propanediol	57-55-6	C_3_H_8_O_2_	Aladdin	>0.99

**Table 2 polymers-14-01783-t002:** Betaine-based DESs prepared in this work.

HBA	HBD	Mole Ratio	Solutions
Betaine	1,2-Propanediol	1:3.5	DES1
		1:4	DES2
		1:5	DES3
		1:6	DES4

**Table 3 polymers-14-01783-t003:** Density of 1,2-propanediol and DESs in different temperatures *.

T/(K)	*ρ*_exp_/(kg·m^−^^3^)				
	1,2-Propanediol	DES1	DES2	DES3	DES4
293.15	1.0362	1.0756	1.0727	1.0670	1.0629
298.15	1.0326	1.07220	1.0694	1.0636	1.0595
303.15	1.0288	1.0687	1.0661	1.0602	1.0561
308.15	1.0251	1.0652	1.0628	1.0568	1.0527
313.15	1.0213	1.0617	1.0594	1.0534	1.0492
318.15	1.0175	1.0581	1.0561	1.0500	1.0457
323.15	1.0136	1.0546	1.0527	1.0466	1.0422
328.15	1.0098	1.0512	1.0493	1.0431	1.0387
333.15	1.0058	1.0478	1.0459	1.0396	1.0352
338.15	1.0018	1.0446	1.0425	1.0361	1.0316
343.15	0.9978	1.0415	1.0391	1.0326	1.0281
348.15	0.9937	1.0385	1.0357	1.0291	1.0245
353.15	0.9897	1.0354	1.0322	1.0255	1.0208
358.15	0.9855	1.0321	1.0287	1.0220	1.0172
363.15	0.9813	1.0286	1.0252	1.0184	1.0135

* The standard uncertainties (*u*) are *u*(*T*) = 0.01 K and *u_r_*(*ρ*) = 0.001.

**Table 4 polymers-14-01783-t004:** Parameters in equation (1) for 1,2-propanediol and different DESs.

Parameters	1,2-Propanediol	DES1	DES2	DES3	DES4
*A*/(kg·m^−3^)	1.26697	1.27121	1.27177	1.27062	1.26997
*B* × 10^−4^/(kg·m^−3^·K^−1^)	−7.8514	−6.68825	−6.78264	−6.93838	−7.05260

**Table 5 polymers-14-01783-t005:** Surface tension of 1,2-propanediol and different DESs *.

*T*/(K)	*σ*_exp_/(mN·m^−1^)				
	1,2-Propanediol	DES1	DES2	DES3	DES4
293.15	35.92	51.26	46.10	41.61	45.34
313.15	35.16	49.98	44.70	41.09	44.01
333.15	34.40	49.60	44.38	40.69	43.42
353.15	33.64	48.67	42.68	39.98	42.82

* The combined expanded uncertainties *U*_c_ are *U*_c_(*T*) = 0.02 K, *U*_c_(*σ*) = 0.02 in the level of confidence 0.95.

**Table 6 polymers-14-01783-t006:** Fitted coefficients of surface tension of 1,2-propanediol and different DESs.

Parameters	1,2-Propanediol	DES1	DES2	DES3	DES4
*a*/(mN·m^−1^)	47.0597	63.0459	61.5596	49.3898	57.0659
*b*/(mN·m^−1^·K^−1^)	−0.0380	−0.0408	−0.0529	−0.0265	−0.0408

**Table 7 polymers-14-01783-t007:** Viscosity of 1,2-propanediol and different DESs *.

*T*/(K)	*η*_exp_/(mPa·s)				
	1,2-Propanediol	DES1	DES2	DES3	DES4
293.15	56.876	178.952	156.418	132.340	116.689
303.15	33.897	103.477	84.311	71.552	62.524
313.15	19.980	57.383	49.716	41.169	36.191
323.15	12.956	34.602	30.431	25.160	22.495
333.15	8.399	23.351	20.605	17.330	15.961
343.15	5.466	16.862	15.096	12.929	11.791
353.15	3.721	12.785	11.373	9.920	9.017

* The combined expanded uncertainties *U*_c_ are *U*_c_(*T*) = 0.02 K, *U*_c_(*η*) = 0.03 in the level of confidence 0.95.

**Table 8 polymers-14-01783-t008:** Parameters in Equation (3) for different DESs.

DESs	*η*_0_ /(mPa·s)	*E_η_*/(kJ·mol^−1^)	*r* ^2^
DES1	2.3098 × 10^−5^	38.49	0.9929
DES2	2.7029 × 10^−5^	37.70	0.9926
DES3	2.6080 × 10^−5^	37.35	0.9894
DES4	3.0196 × 10^−5^	36.66	0.9889

**Table 9 polymers-14-01783-t009:** Comparison between experimental and calculated viscosity of different DESs.

*T*/(K)	293.15	303.15	313.15	323.15	333.15	343.15	353.15
**DES1** *E_η_ =* 38.49 (kJ·mol^−1^) *η*_0_ = 2.3098 × 10^−5^ (mPa·s)							
*η*_exp_/(mPa·s)	178.952	103.477	57.383	34.602	23.351	16.862	12.785
*η*_cal_/(mPa·s)	166.733	99.033	60.812	38.486	25.035	16.698	11.396
Dev(%)	6.82	4.29	5.97	11.22	7.21	0.97	10.86
**DES2** *E_η_* = 37.70 (kJ·mol^−1^) *η*_0_ = 2.7029 × 10^−5^ (mPa·s)							
*η*_exp_/(mPa·s)	156.418	84.311	49.716	30.431	20.605	15.096	11.373
*η*_cal_/(mPa·s)	141.128	84.726	52.549	33.571	22.031	14.817	10.192
Dev(%)	9.77	0.49	5.70	10.32	6.92	1.85	10.38
**DES3** *E_η_* = 37.35 (kJ·mol^−1^) *η*_0_ = 2.6080 × 10^−5^ (mPa·s)							
*η*_exp_/(mPa·s)	132.340	71.552	41.169	25.160	17.330	12.929	9.920
*η*_cal_/(mPa·s)	117.957	71.151	44.326	28.436	18.734	12.647	8.729
Dev(%)	10.87	0.56	7.67	13.02	8.10	2.18	12.00
**DES4** *E_η_* = 36.66 (kJ·mol^−1^) *η*_0_ = 3.0196 × 10^−5^ (mPa·s)							
*η*_exp_/(mPa·s)	116.689	62.524	36.191	22.495	15.961	11.791	9.017
*η*_cal_/(mPa·s)	102.901	62.652	39.374	25.467	16.908	11.497	7.990
Dev(%)	11.82	0.20	8.80	13.21	5.93	2.49	11.39

## Data Availability

Not applicable.

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
