# Peer review of "Physical Properties of Betaine-1,2-Propanediol-Based Deep Eutectic Solvents"

_polymers, 2022, doi:10.3390/polym14091783_

Round 1

Reviewer 1 Report

The paper reports about carful measurements on Deep Eutectic Solvents and provides valuable conclusions. The fig caption for Fig 1 should read two solvents not three. Otherwise the paper can be accepted for publication.

Author Response

We thank the reviewer for the carefully review. The error has been revised in Figure 1.

Reviewer 2 Report

Physical Properties of Betaine-1,2-propanediol Based Deep Eutectic Solvents
Manuscript ID: polymers- 1630108
Comments to the Authors

This paper reports the physical properties (density, surface tension, and viscosity) of four deep eutectic solvents (DESs) synthesized by blending betaine as hydrogen bond acceptor and 1,2-propanediol as hydrogen bond donor in different molar ratios. The authors report the effect of molar ratios on the physical properties and also report the empirical relationships for the dependence of the physical properties (density, surface tension, and viscosity) on temperature.

Although the applications of betaine based DESs have been reported, their physical properties have been rarely explored. The significance of this paper is that it reports the physical properties of betaine based DESs and their dependence on molar ratio and temperature. The knowledge of physical properties will help design optimal systems for industrial applications and also in elucidating the synthesis mechanism and microstructure.

The paper is technically sound and written in an easy-to-understand style. The authors provide a good background and introduction to the study. They have presented the technical terms and explained the experimental results in an easy-to-understand manner. No further experiments are needed, but I think this manuscript can potentially be improved through rephrasing some of the paragraphs and re-organization of some figures and sections. I have some comments and suggestions as listed below:

  1. The authors have reported the comparison between the physical properties of betaine based DESs and pure 1,2-propanediol qualitatively. Can they also comment on comparison with properties of pure betaine?
  2. It might also be useful to discuss the comparisons more quantitatively, such as adding the values for the physical properties of pure components in the same plots as the DESs.
  3. The densities have been reported for temperature range 293.15 K to 363.15 K, while the surface tension and viscosity have been reported for 293.15 K to 353.15 K. Can the authors comment on why different temperature ranges were chosen?
  4. Line 108: The authors mention that homogeneous mixtures could not be formed for HBA: HBD ratios 1:2 and 1:3. It might be helpful to explain why that is the case.
  5. Most of the values for physical properties have been reported with large no. of significant digits. It might be easier to read if the significant digits are reduced, say to 4 or 5.
  6. Line 196: The authors mention that addition of betaine increases the surface tension of DES, in contrast to choline chloride. Can the authors explain this in more detail?
  7. Figure 4: The surface tension decreases from DES1 to DES3 and increases from DES3 to DES4. The author cites optimal hydrogen bonding as the reason behind this behaviour. It might be helpful if the authors can explain the mechanism behind the optimal hydrogen bonding at a certain molar ratio.
  8. The introduction section seems a little wordy and the readability might be improved by use of a more concise language.
  9. In abstract, the acronym DES might need to be spelled out in first mention.
  10. Lines 62-83: The authors have provided several references for applications of betaine based DESs. But the flow of the literature review can be further improved by tying them in a more coherent manner.

Author Response

Please find the point by point responses in the attachment.

Round 2

Reviewer 1 Report

The revised version is fine and the article should now be accepted for publication

Author Response

Thanks for your review and recognition of our work

Reviewer 2 Report

 I think the authors have responded well to my comments. I just have two more additional minor comments: 1. The authors responded that they are using molecular dynamic simulations to answer some unanswered questions in this research. It might be helpful that they briefly discuss this somewhere in the discussion or conclusion. Like, summarizing the not yet answered questions and how they might need MD simulations to answer those. 2. Some spell checks and grammatical checks might be required.

Author Response

Please find it in the attachment.

Round 3

Reviewer 2 Report

I find all my comments addressed.